# Dietary Daily Sodium Intake Lower than 1500 mg Is Associated with Inadequately Low Intake of Calorie, Protein, Iron, Zinc and Vitamin B1 in Patients on Chronic Hemodialysis

**DOI:** 10.3390/nu12010260

**Published:** 2020-01-19

**Authors:** Maurizio Bossola, Enrico Di Stasio, Antonella Viola, Stefano Cenerelli, Alessandra Leo, Stefano Santarelli, Tania Monteburini

**Affiliations:** 1Servizio Emodialisi, Università Cattolica del Sacro Cuore di Roma, Fondazione Policlinico Agostino Gemelli, IRCCS, 00168 Roma, Italy; 2UOC Chimica, Università Cattolica del Sacro Cuore, Biochimica e Biologia Molecolare Clinica, Fondazione Policlinico Universitario A. Gemelli, IRCCS, 00168 Roma, Italy; enrico.distasio@policlinicogemelli.it; 3Servizio Nutrizione Clinica, Università Cattolica del Sacro Cuore di Roma, Fondazione Policlinico Agostino Gemelli, IRCCS, 00168 Roma, Italy; dott.ssa.antonellaviola@gmail.com (A.V.); alessandra.leo@gmail.com (A.L.); 4Unità Operativa Nefrologia ed Emodialisi, Ospedale “Principe di Piemonte”, 60019 Senigallia, Italy; stefano.cenerelli@sanita.marche.it; 5Unità Operativa Nefrologia ed Emodialisi, Ospedale “A. Murri”, 60035 Jesi, Italy; stefano.santarelli@sanita.marche.it (S.S.); tania.monteburini@sanita.marche.it (T.M.)

**Keywords:** sodium, salt, hemodialysis, dietary intake, calorie, energy, protein, vitamin, trace element

## Abstract

**Background:** To measure daily sodium intake in patients on chronic hemodialysis and to compare the intake of nutrients, minerals, trace elements, and vitamins in patients who had a daily sodium intake below or above the value of 1500 mg recommended by the American Heart Association. **Methods:** Dietary intake was recorded for 3 days by means of 3-day diet diaries in prevalent patients on chronic hemodialysis. Each patient was instructed by a dietitian on how to fill the diary, which was subsequently signed by a next of kin. **Results:** We studied 127 patients. Mean sodium intake (mg) was 1295.9 ± 812.3. Eighty-seven (68.5%) patients had a daily sodium intake <1500 mg (group 1) and 40 (31.5%) ≥ 1500 mg (group 2). Correlation between daily sodium intake and daily calorie intake was significant (*r* = 0.474 [0.327 to 0.599]; *p* < 0.0001). Daily calorie intake (kcal/kg/day) was lower in group 1 (21.1 ± 6.6; *p* = 0.0001) than in group 2 (27.1 ± 10.4). Correlation between daily sodium intake and daily protein intake was significant (*r* = 0.530 [0.392 to 0.644]; *p* < 0.0001). The daily protein intake (grams/kg/day) was lower in group 1 (0.823 ± 0.275; *p* = 0.0003) than in group 2 (1.061 ± 0.419). Daily intake of magnesium, copper, iron, zinc, and selenium was significantly lower in group 1 than in group 2. Daily intake of vitamin A, B2, B3, and C did not differ significantly between group 1 and group 2. Daily intake of vitamin B1 was significantly lower in group 1 than in group 2. Significantly lower was, in group 1 than in group 2, the percentage of patients within the target value with regard to intake of calories (11.5% vs. 37.5%; *p* = 0.001) and proteins (9.2% vs. 27.5%; *p* = 0.015) as well as of iron (23% vs. 45%; *p* = 0.020), zinc (13.8% vs. 53.8%; *p* = 0.008) and vitamin B1 (8.1% vs. 50%; *p* < 0.001). **Conclusion:** A low daily intake of sodium is associated with an inadequately low intake of calorie, proteins, minerals, trace elements, and vitamin B1. Nutritional counselling aimed to reduce the intake of sodium in patients on chronic hemodialysis should not disregard an adequate intake of macro- and micronutrients, otherwise the risk of malnutrition is high.

## 1. Introduction

Excess sodium intake is related to high blood pressure, cardiovascular disease and stroke in the general population [1,2]. In patients on chronic hemodialysis, excess sodium intake is also associated to thirst and consequently to high interdialytic weight gain (IDWG). High IDWG is associated with higher risk of all-cause and cardiovascular death and increased morbidity, such as ventricular hypertrophy and major adverse cardiac and cerebrovascular events [3,4,5,6,7,8,9]. In addition, it leads to supplementary weekly dialysis sessions with consequent deterioration of quality of life and increased costs [10].

The American Heart Association recommends a daily sodium intake <1500 mg [11]. The average daily sodium intake among dialysis patients has been reported to be significantly higher [12,13,14,15,16,17,18]. Sodium intake of patients on chronic hemodialysis varies according to the country, being higher especially where the diet is rich in processed foods [12,13,14,15,16,17].

Nephrologists routinely recommend restriction of salt, but unfortunately it is difficult to obtain in daily clinical practice. Adherence to a low-salt diet of hemodialysis patients is poor [18] as well as in patients with other chronic diseases such as heart failure, hypertension, and cirrhosis [19,20,21,22,23,24,25,26,27]. Many factors contribute to such poor adherence such as lack of knowledge, interference with socialization, low education level, low socioeconomic status, and lack of food selections [10]. It must also be considered that hemodialysis patients are continuously instructed to follow a restricted diet because of potassium and phosphorus concerns, and such further restrictions may lead to a diet poorly acceptable in terms of palatability and pleasantness.

Recently, it has been suggested that that sodium restriction may be beneficial as long as optimal nutritional status and food intake are not compromised [28]. Indeed, malnutrition and the risk of protein-energy wasting may occur following a sodium-restricted diet because sodium is widely used in the foods [28]. It has also been suggested that the efforts to intensify sodium restriction might increase the risk of compromising energy intake further [17].

The present study aims to compare the intake of nutrients, minerals, trace elements, and vitamins in patients who had a daily sodium intake below or above the value of 1500 mg recommended by the American Heart Association.

## 2. Patients and Methods

The study was performed at three centers of our country. Exclusion criteria included: age <18 years, dialytic age ≤6 months, inability to answer to the questionnaires, and diagnosis of anorexia nervosa. None of the patients included in the study were receiving appetite stimulants, nutrition supplements or advise to increase nutrient intake. The study was carried out following the rules of the Declaration of Helsinki of 1975. The study was approved by the local ethic committee (P/608/CE/2011; 01/08/2011) and written informed consent was obtained from all patients before enrollment in the study.

### 2.1. Hemodialysis

HD patients were maintained on regular HD prescription, three times a week, for 4 h per session. The blood flow ranged from 250 to 300 mL/min with a dialysis rate flow of 500 mL/min. All patients were treated with high-permeability membranes.

### 2.2. Assessment of Dietary Intake

Dietary intake was recorded for 3 days by means of 3-day diet diaries. Each patient was instructed by a dietitian on how to fill the diary, which was subsequently signed by a next of kin. Dietary records were elaborated by a computerized software (Progeo, Ascoli Piceno, Italy). Data entry and calculation was performed by a dietician (AV). Macro- and micronutrients intakes of the 3 days were calculated and then averaged. American Heart Association recommends a daily sodium intake <1500 mg/day [11]. Thus, we divided patients into two groups according to the daily sodium intake: group 1 < 1500 mg; 2 ≥ 1500 mg.

### 2.3. Statistical Analysis

Statistical analysis was performed by SYSTAT 7.0 software (SPSS, Chicago, IL, USA). All data were expressed as mean ± SD unless otherwise specified. All data were first analyzed for normality of distribution using the Kolmogorov-Smirnov test of normality. When comparing differences in the groups, the Kruskal-Wallis non-parametric test was used for non-normally distributed continues variables, and analysis of variance was used for normally distributed variables. *p* value of less than 0.05 was considered statistically significant.

## 3. Results

One-hundred-and-twenty-seven patients were included in the study. Their demographic, clinical and laboratory characteristics as well as the causes of ESRD are shown in Table 1.

The mean sodium intake (mg) was 1295.9 ± 812.3. The mean sodium intake per body weight was 20.1 ± 14 mg/kg. Eighty seven (68.5%) patients had a daily sodium intake <1500 mg (group 1) and 40 (31.5%) ≥ 1500 mg (group 2).

The mean sodium intake was similar in dialysis and non-dialysis days. The correlation between sodium intake and age was negative and statistically significant (*r* = −0.203 [−0.365–0.029]; *p* < 0.0001). The mean age of patients of group 1 and 2 did not differ significantly (Table 2).

The daily sodium intake was significantly higher in patients <40 years (2324 ± 1666) than in patients of 40–60 years (1319 ± 475) or older (1220 ± 787) (*p* = 0.005). However, the difference between the latter two groups was not statistically significant.

The correlation between daily sodium intake and daily calorie intake was highly significant (Figure 1).

The daily calorie intake was significantly lower in group 1 than in group 2 (*p* = 0.0001). (Table 2). Similarly, the correlation between daily sodium intake and daily protein intake was highly significant (Figure 2).

The daily protein intake was significantly lower in group 1 than in group 2 (*p* = 0.0003) (Table 2).

The correlation between daily sodium intake and daily carbohydrate intake was highly significant (*r* = 0.415 [0.260–0.549]: *p* < 0.0001). The daily carbohydrate intake was significantly lower in group 1 than in group 2 (Table 2). The correlation between daily sodium intake and daily lipid intake was highly significant (*r* = 0.421 [0.267–0.555]; *p* < 0.0001). The daily lipid intake was significantly lower in group 1 than in group 2 (Table 2).

The correlation between daily sodium intake and daily phosphate intake was highly significant (Figure 3). The daily intake of phosphate was significantly higher in group 2 than in group 1 (Table 2). The correlation between daily sodium intake and daily potassium intake was highly significant. The daily intake of potassium was significantly higher in group 2 than in group 1 (Table 2).

The correlations between daily sodium intake and daily intake of magnesium, copper, iron, zinc, and selenium were significant (data not shown). The daily intake of magnesium, copper, iron, zinc, and selenium was significantly lower in group 1 than in group 2 (Table 3).

The daily intake of vitamin A, B2, B3, and C did not differ significantly between group 1 and group 2. The correlation between daily sodium intake and daily vitamin B1 intake was highly significant (*r =* 0.763 [0.679 to 0.827]; *p* < 0.0001) (Figure 3).

The daily intake of vitamin B1 was significantly lower in group 1 than in group 2 (Table 4).

We then compared patients of group 1 and group 2 in terms of percentage of patients within the recommended value of nutrients intakes. The percentage of patients with intakes of calories and proteins (Table 5) target value were significantly lower in group 1 than in group 2.

Similarly, the percentage of patients with intakes of iron and zinc (Table 6) and vitamin B1 (Table 7) within the target value were significantly lower in group 1 than in group 2. Conversely, the percentage of patients with adequate potassium intake was significantly higher in group 1 than in group 2 (Table 5).

## 4. Discussion

More than two thirds of patients included in the present study had a sodium intake <1500 mg/day, and thus were within the value recommended by the American Heart Association [11]. It is possible that these good results are a consequence of the habit of the patients of our country to consume fresh foods and avoid, as much as possible, the use of fast and/or processed foods. The average daily salt intake among dialysis patients has been reported to be high, ranging from 7.9 to 14.1 g/day [12,13,14,15,16,17]. Recently, Luis et al., that referred to the recommendations of 2000–2300 mg/day of the European Best Practice Guideline on Nutrition and Chronic Kidney Disease, found that only 15% of patients were within the target value [29]. Similarly, in the study of Xie et al., more than half of the patients exceeded the recommended sodium intake of 1840–2300 mg/day of the New Zealand Dietitians Guidelines [17].

In patients on chronic hemodialysis, a diet with a low sodium intake is associated with an improvement of xerostomia, a better control of blood pressure and with a lower interdialytic weight gain [35,36], as well as with a reduction of the risk of all-cause and cardiovascular death and morbidity, such as ventricular hypertrophy and major adverse cardiac and cerebrovascular events [3,4,5,6,7,8,9].

However, the present study also shows that patients with a daily sodium intake lower than 1500 mg, with respect to those with an intake ≥1500 mg, had a significantly greater risk to have an inadequate calorie and protein intake as well as of a low intake of iron and zinc. This in accordance with the study of Xie et al. who appropriately suggested that the efforts to intensify sodium restriction might increase the risk of compromising energy intake further [17]. Inadequate low calorie and protein intakes put the patients at risk of protein energy wasting and to its consequences. However, the results of the present study do not mean that patients should increase their calorie and protein intakes while disregarding the daily sodium intake. Instead, our study suggests that much attention should be paid to the nutritional education of patients on chronic hemodialysis with the aim to stimulate them to increase their food intake and, at the same time, limit as much as possible the daily sodium consumption. Undoubtedly, this is not an easy task. Adherence to a low-salt diet of hemodialysis patients is poor [18,19] as well as in patients with other chronic diseases [20,21,22,23,24,25,26,27,28], even though a high percentage of patients on hemodialysis recognize that salty food is not good for them [18,37,38]. Sevick et al. showed that nutritional counseling and a technology-supported behavioral intervention resulted in reduced dietary sodium intake at 8 weeks, however, it was not sustained at 16 weeks [39]. Conversely, 48-month nutritional counseling resulted in a significant decrease of salt and water intake (from 13.3 ± 2.7 to 11.8 ± 2.4 g/day and 2528 ± 455 to 2332 ± 410 mL/day, respectively) [39]. In addition, it is usually difficult to combine an adequate calorie and protein intake with a low sodium intake. Interestingly, in all the studies cited above the intake of macronutrients and micronutrients was never reported. Nevertheless, in the randomized, controlled study by Rodrigues-Telini, sodium intake was reduced by 2 g/day for 16 weeks without a reduction in total caloric or protein intake as a consequence of adequate nutritional counselling [39].

The lower percentage of patients within the recommended value of iron and zinc intake, observed in the group with a sodium intake <1500 mg, requires attention also. Iron is essential for erythropoiesis, and zinc deficiency is associated with delayed wound healing, immune deficiency, dysgeusia, decreased cognitive function, anorexia, and increased risk of infection [32,40]. Attention should be paid to stimulate the patients to consume food rich in zinc (such as bread, cereals, leguminous plant seeds and mushrooms, red meat) and in iron (red meat, chicken, turkey, pork, ham, veal, fish, raw, and cooked spinach).

The percentage of patients within the target value of vitamins intake was low both in patients with daily sodium intake <1500 mg and in those with an intake ≥1500. However, the difference between the two groups was highly significant with regard to vitamin B1 intake, with only 8% and 50% of patients with a sodium intake <1500 or ≥1500 mg within the target value, respectively. Low vitamin B1 intake can lead to low serum levels and associated complications as Wernicke’s encephalopathy [41,42].

Overall, the correlation between sodium intake and age was significant and the daily sodium intake was significantly higher in patients <40 years than in patients of 40–60 years or older, although the calorie intake was similar in the three groups. Similar observations have been reported by other authors [19]. Conversely, in the study of Gkza et al., older patients consumed more sodium than the middle-age cohort and this was due, according to the authors, to the fact that many elderly patients that were no longer able to cook fresh food increasingly relied on ready and prepared meals that have higher sodium content and also because elderly patients residing in nursing or residential homes did not receive low-sodium meals [16]. In the present study, all old patients lived at home and consumed food prepared by themselves or by relatives. Overall, the observation that younger patients consume more sodium may be useful in the clinical practice because it allows to identify a specific population at risk of high sodium intake that needs to be offered adequate nutritional counselling.

The present study has a limitation. We used a 3-day food diary to assess sodium intake, a method that may imply inherent errors of dietary assessment and can be the cause of under- and over-reporting. Nevertheless, there is large recent evidence that the 3-day diet diary is a simple, useful, and effective method to measure the intake of macro- and micro-nutrients in patients on chronic hemodialysis [43,44,45,46,47]. The strength is that we included patients of different towns of Italy and consequently the data can be considered representative of the Italian population of patients on chronic hemodialysis.

In conclusion, the results of the present study suggests practically that low sodium intake is associated with a low intake of calorie, protein, iron, zinc, and vitamin B. This finding implies and strongly suggests that all nutritional counselling aimed at reducing the daily intake of sodium in patients on chronic hemodialysis should not disregard an adequate intake of macro- and micronutrients, otherwise the risk of protein and energy wasting as well as the deficit of some trace elements and vitamins may be high.

## Figures and Tables

**Figure 1 nutrients-12-00260-f001:**
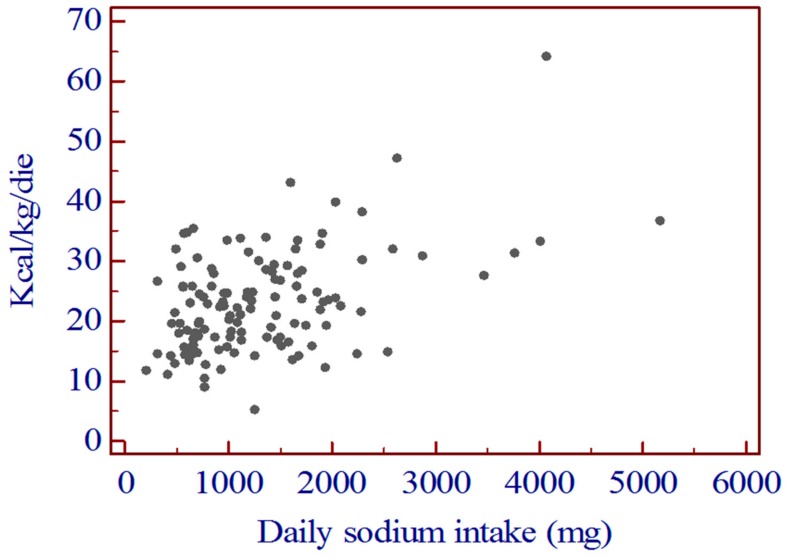
Correlation between daily sodium intake and daily calorie intake. *r* = 0.474 [0.327–0.599]; *p* < 0.0001.

**Figure 2 nutrients-12-00260-f002:**
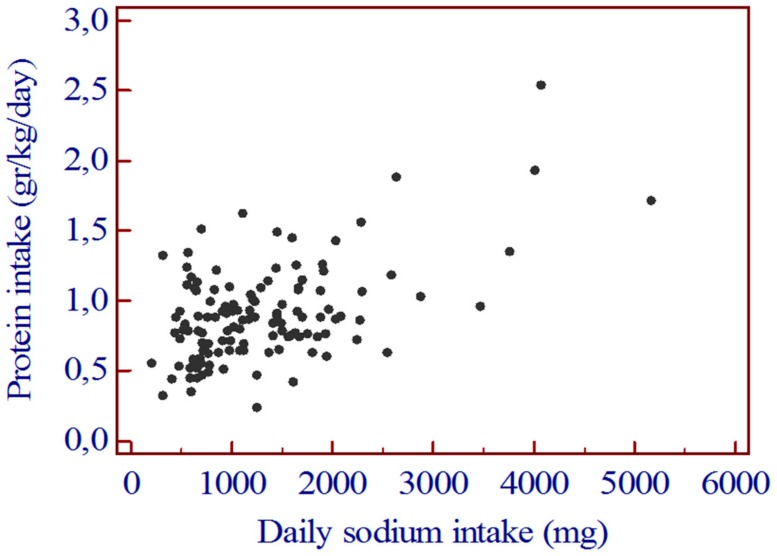
Correlation between daily sodium intake and daily protein calorie intake. *r* = 0.530 [0.392–0.644]; *p* < 0.0001.

**Figure 3 nutrients-12-00260-f003:**
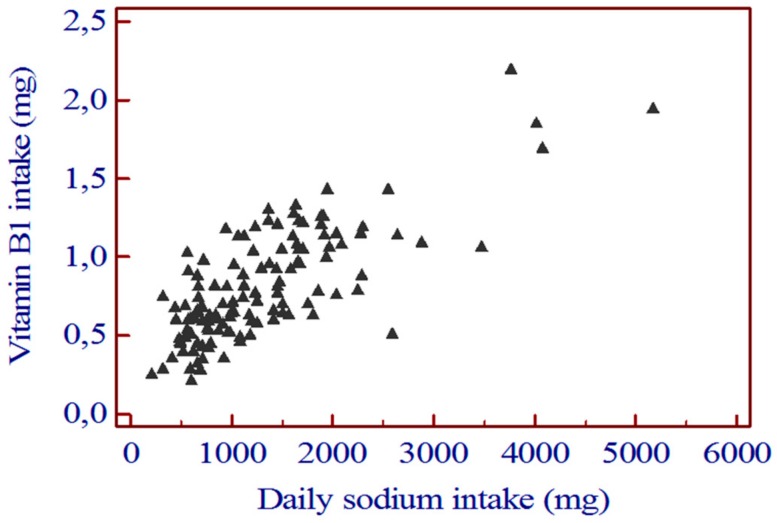
Correlation between daily sodium intake and daily vitamin B1. *r* = 0.763 [0.679 to 0.827]; *p* < 0.0001.

**Table 1 nutrients-12-00260-t001:** Characteristics of 127 patients included in the study.

Characteristics	
Number of patients	127
Age (yrs)	69.8 ± 13.8
Males: Females	75:52
Primary cause of ESRD:	
-hypertension	41 (32.6)
-glomerulonephritis	30 (23.4)
-diabetes	27 (21.2)
-interstitial nephritis	22 (20.3)
-polycystic renal disease	5 (3.9)
-others/unknown	2 (1.5)
Duration of dialysis (yrs)	6.4 ± 4.9
Body Mass Index	23.8 ± 5.1
Vascular access	
Arterovenous fistula	108 (85%)
Central venous catheter	19 (15%)
Urea, mg/dL	82.2 ± 14.5
Creatinine, mg/dL	10.2 ± 2.8
Kt/V	1.28 ± 1.2

**Table 2 nutrients-12-00260-t002:** Daily intake of calorie, protein, and fat according to the daily sodium intake.

	Daily Sodium Intake (mg)		
	<1500 (*n* = 87)	≥1500 (*n =* 40)	*p*
Age (years)	71.3 ± 13.3	66.8 ± 14.7	0.088
Sex (M:F)	51:36	24:16	1.000
BMI	23.7 ± 4.5	24.3 ± 5.2	0.265
Charlson Comorbidity Index	1.5 ± 1.4	1.6 ± 1.3	0.347
Calorie (kcal/kg/day)	21.1 ± 6.6	27.1 ± 10.4	0.0001
Protein (g/kg/day)	0.823 ± 0.275	1.061 ± 0.419	0.0003
Protein, % TE	16 ± 3.6	16.1 ± 3.4	0.864
Lipid (grams)	47.1 ± 15	56.1 ± 20.6	0.006
Lipid. % TE	32.7 ± 5.4	30.2 ± 5.9	0.027
Carbohydrate (grams)	186.8 ± 57.5.1	256.5 ± 87.4	<0.0001
Carbohydrate, % TE	51.3 ± 6.9	53.6 ± 8.1	0.104
Cholesterol (mg)	182.4 ± 90.8	232.6 ± 129	0.013
Saturated fat (grams)	5.1 ± 2.4	6.3 ± 2.3	0.009
MUFA (grams)	27.2 ± 6.1	32.4 ± 8.8	0.0001
PUFA (grams)	2.71 ± 1.15	4.23 ± 1.50	<0.0001
Omega-6 FA (grams)	4.57 ± 1.26	5.87 ± 1.63	<0.0001
Omega-3 FA (grams)	0.451 ± 0.188	0.591 ± 0.219	0.0003
Fiber (grams)	10.9 ± 4.3	12.4 ± 4.4	0.070

**Table 3 nutrients-12-00260-t003:** Daily dietary intake (mean ± SD) of minerals and trace elements according to the daily sodium intake.

	Daily Sodium Intake (mg)		
	<1500 (*n =* 87)	≥1500 (*n* = 40)	*p*
Phosphorus (mg)	749.4 ± 213.4	910.2 ± 265	0.0004
Potassium (mg)	1461.1 ± 469	1779 ± 583	0.001
Calcium (mg)	330 ± 155.9	386.6 ± 171.4	0.068
Magnesium (mg)	158.6 ± 46.3	188.4 ± 54.9	0.001
Copper (mg)	31.5 ± 18.2	68.6 ± 36.1	<0.0001
Selenium (mg)	42.8 ± 13.4	68.1 ± 23.7	<0.0001
Iron (mg)	6.56 ± 1.97	7.81 ± 2.58	0.003
Zinc (mg)	6.73 ± 2.12	8.39 ± 2.65	0.0003

**Table 4 nutrients-12-00260-t004:** Daily dietary intake (mean ± SD) of vitamins according to the daily sodium intake.

	Daily Sodium Intake (mg)		
	<1500 (*n* = 87)	≥1500 (*n* = 40)	*p*
Vitamin A (μg)	437.7 ± 214.1	460.5 ± 251.1	0.233
Vitamin B1 (mg)	0.68 ± 0.24	1.12 ± 0.35	<0.0001
Vitamin B2 (mg)	1.02 ± 0.30	1.13 ± 0.38	0.087
Vitamin B3 (mg)	12.3 ± 4.7	13.8 ± 4.4	0.098
Vitamin C (mg)	44.5 ± 30.3	48.2 ± 47.2	0.599
Vitamin E (mg)	9.1 ± 2.3	9.5 ± 2.6	0.407

**Table 5 nutrients-12-00260-t005:** Recommended daily dietary intake (mean ± SD) of nutrients and fibers and percentage of patients within the target value, according to the daily sodium intake (Group 1: <1500 mg; Group 2: ≥1500 mg).

		Percentage of Patients within Target Values		
	Current Recommendations	Group 1	Group 2	*p*
Calorie *	30–35 kcal/kg/day	11.5	37.5	0.001
Protein *	1–1.2 g/kg/day	9.2	27.5	0.015
Carbohydrates	45–65% energy	81.6	90	0.298
Lipid **	20–35% energy	71.3	82.5	0.265
Cholesterol **	<200 mg/day	61.6	52.5	0.438
Omega-3 FA **	1–1.5 g/day	2.3	5.3	0.589
Omega-6 FA **	4.5–6 g/day	43.7	27.5	0.115
Fiber #	>25 g/day	2.3	10	0.147

* Target values recommended by KDOQI Clinical Practice Guidelines for Nutrition in Chronic Renal Failure [29]; ** Target values recommended by KDOQI Clinical Practice Guidelines for Management of Dyslipidemias in Patients with Kidney Disease [30]; # Target values recommended by Standing Committee on the Scientific Evaluation of Dietary Reference Intakes FaNB [31].

**Table 6 nutrients-12-00260-t006:** Recommended daily dietary intake (mean ± SD) of minerals and trace elements and percentage of patients within the target value, according to the daily sodium intake (Group 1: <1500 mg; Group 2: ≥1500 mg). M, men; W, women.

		Percentage of Patients within Target Values		
	Current Recommendations	Group 1	Group 2	*p*
Phosphorus *	<0.8 g/day	55.2	35	0.054
Magnesium *	300–400 mg/day	0	0	1.000
Calcium *	<2 g/day	100	100	1.000
Potassium *	1950–2730 mg/day	100	92.5	0.022
Iron *	8 mg (M)/15 mg (W)/day	23	45	0.020
Zinc *	10–15 mg (M)/8–12 mg (W)	13.8	53.8	0.008

* Target values recommended by European Best Practice Guideline on Nutrition and Chronic Kidney Disease [32].

**Table 7 nutrients-12-00260-t007:** Recommended daily dietary intake (mean ± SD) of vitamins and percentage of patients within the target value, according to the daily sodium intake (Group 1: <1500 mg; Group 2: ≥1500 mg). M, men; W, women.

		Percentage of Patients within Target Values		
	Current Recommendations	Group 1	Group 2	*p*
Vitamin A (μg) *	≥900 μg (M)/≥700 μg (W)	3.4	5	0.941
Vitamin B1 (mg) *	≥1.3 mg (M)/≥1.1 mg (W)	8.1	50	<0.001
Vitamin B2 (mg) *	1.3–1.7 mg	18.3	30	0.216
Vitamin B3 (mg) *	≥16 mg (M)/≥14 mg (W)	37.9	42.5	0.768
Vitamin C (mg) *	≥90 mg (M)/≥75 mg (W)	12.6	20	0.416
Vitamin E (mg) *	≥15 mg	0	5	0.181

* Standing Committee on the Scientific Evaluation of Dietary Reference Intakes FaNB Thiamin, Riboflavin, Niacin, Vitamin B6, Folate, Vitamin B12, Pantothenic Acid, Biotin, and Choline. 1998. Washington, DC: National Academies Press; 1998 [33,34].

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
