# Peer review of "Dietary Daily Sodium Intake Lower than 1500 mg Is Associated with Inadequately Low Intake of Calorie, Protein, Iron, Zinc and Vitamin B1 in Patients on Chronic Hemodialysis"

_nutrients, 2020, doi:10.3390/nu12010260_

Round 1
Reviewer 1 Report
Was the sample size adequate for this study?
Was the salt intake low because the cases were not eating adequately and not the other way around?
The limitation as pointed out by the authors may be major enough to skew the results.
There are minor grammatical errors that need to be reviewed.
Author Response
Was the sample size adequate for this study? Yes, it was. On the basis of our sample size and an a=0.05 a power of 90.7% was estimated.
Was the salt intake low because the cases were not eating adequately and not the other way around? Indeed, in the routine clinical practice, we recommend to strongly limit the intake of salt (essentially with the aim to control blood pressure and, above all, to reduce thirst and limit the interdialytic weight gain). Nevertheless, we do not recommend to limit the calorie and protein intakes in order to avoid malnutrition and, especially in older patients, sarcopenia. However, it is possible that hemodialysis patients, being frightened by the high interdialytic weight gain, can highly limit their salt intake and to obtain this may limit calorie and protein intake. Accordingly, the final message of this study is that, because of the large dietary restrictions, hemodialysis patients “suffer” an inadequate diet in the routine clinical practice, especially when the salt intake is reduced. However, the results of the present study do not mean that patients should increase their calorie and protein intakes disregarding of the daily sodium intake. Instead, our study suggests that much attention should be paid to the nutritional education of patients on chronic hemodialysis with the aim to stimulate them to increase their food intake and, at the same time, limit as much as possible the daily sodium consumption
The limitation as pointed out by the authors may be major enough to skew the results. Indeed, there is large evidence, also recent, that the 3-day diet diary is a simple, useful, and effective method to measure the intake of macro- and micro-nutrients in patients on chronic hemodialysis (references 44-48). Accordingly, a recent study, published in Nutrients and aimed at evaluating “Dietary Sodium and Other Nutrient Intakes among Patients Undergoing Hemodialysis in New Zealand” (Nutrients 2018;10:502), has used also a 3-day diet diary.
There are minor grammatical errors that need to be reviewed. We did this.
Reviewer 2 Report
The phrase "was significantly" is doubled in line 128, 134 - please correct
There are several typos and grammatical errors fe. in the description of Table 6
When describing the statistical significance between groups please provide p value fe. line 128, 134
Have you performed the measurements of BMI and other factors before and after the 3 day diet ? The study would benefit if the control group was also selected and described.
Please discuss the duration of the diet and potential results of the long term diet
The reference list seems a little outdated, provide more recent publications fe.
Serum iron, magnesium, copper, and manganese levels in alcoholism: a systematic review. [AUT.] CEZARY GROCHOWSKI, ELIZA BLICHARSKA, JACEK BAJ, ALEKSANDRA MIERZWIŃSKA, KAROLINA BRZOZOWSKA, ALICJA FORMA, RYSZARD MACIEJEWSKI. Molecules [online] 2019 vol. 24 nr 7 [art. nr] 1361, s. 1-14, bibliogr. poz. 75, [przeglądany 8 kwietnia 2019]. Dostępny w: https://www.mdpi.com/1420-3049/24/7/1361. DOI: 10.3390/molecules24071361
Have you tried to do any correlations between trace elements in both groups ?
DId you analyse aluminium intake in those patients ? Hemodialised patients are exposed to high doses of this trace element, which may cause encephalopathy.
Author Response
The phrase "was significantly" is doubled in line 128, 134 - please correct. We have corrected this error
There are several typos and grammatical errors fe. in the description of Table 6. We have corrected these errors and typos
When describing the statistical significance between groups please provide p value fe. line 128, 134. We have added the p values
Have you performed the measurements of BMI and other factors before and after the 3 day diet ? The study would benefit if the control group was also selected and described. Indeed, we did not provide any 3 day diet, in this study. Patients were assuming their regular diet following the recommendations of our nutritional counselling. In this regard, we recommend to limit the intake of potassium, phosphorus, and salt in the clinical practice though a routine nutritional counselling. As consequence, patients on chronic hemodialysis are well aware of the risk associated with an high increase of potassium, phosphorus, and salt. In addition, we monthly measure serum levels of potassium, phosphorus and sodium and through these examinations we can reset the routine nutritional counselling in each patient.
Please discuss the duration of the diet and potential results of the long term diet. In the discussion, we added a paragrapgh in which the advantages of a low-salt diet are described.
The reference list seems a little outdated, provide more recent publications fe. We have provided more recent publications
Have you tried to do any correlations between trace elements in both groups? We found significant correlations, in both groups of salt intake, between intakes of selenium and magnesium, selenium and iron, selenium and zinc, magnesium and iron, magnesium and zinc, and zinc and iron.
Did you analyse aluminium intake in those patients ? Hemodialised patients are exposed to high doses of this trace element, which may cause encephalopathy. Unfortunately, we did not measure aluminium intake in the present study. We are aware that aluminium can induce dementia, osteomalacia, and anemia in patients on chronic hemodialysis. Generally, aluminium present in medications may be ingested but it has been shown that for all drugs consumed, the amount of aluminium ingested versus the blood aluminium level presented no correlation (Bohrer D, et al. Role of medication in the level of aluminium in the blood of chronic haemodialysis patients. Nephrol Dial Transplant. 2009;24:1277–1281)
Round 2
Reviewer 1 Report
Thank you for answering my queries.
Reviewer 2 Report
I recommend this paper for publication.